# Reciprocal Label Diffusion for Learning with Noisy Labels

## Abstract

Deep neural networks are susceptible to overfitting noisy labels, resulting in poor generalization. We propose Reciprocal Label Diffusion (RLD), a novel framework that leverages a mutual guidance mechanism between a label diffusion model and a prediction model to effectively learn from noisy labels. In RLD, the diffusion model is guided by the outputs of the prediction model to denoise corrupted labels through a forward and reverse diffusion process in the logit space, thus modeling and correcting label noise with standard diffusion distributions while enforcing instance-dependency. In turn, the prediction model is refined using the denoised labels produced by the diffusion model, enhancing its learning of accurate representations. This reciprocal interaction enables both models to iteratively enhance each other. To further improve robustness to label noise, we incorporate a contrastive denoising loss that enforces consistency across different data augmentations. Experimental results on benchmark datasets demonstrate that our approach outperforms state-of-the-art methods, achieving significant improvements in classification accuracy under various noise conditions. Our framework provides a robust solution for learning with noisy labels by exploiting the reciprocal interplay between diffusion and prediction models.

## 1 Introduction

Deep neural networks (DNNs) have achieved remarkable success across various domains, including computer vision (Russakovsky et al., 2015; He et al., 2016), natural language processing (Kenton & Toutanova, 2019), and speech recognition (Hinton et al., 2012). This success is largely attributed to their ability to learn complex patterns from large-scale labeled datasets. However, acquiring perfectly labeled data is often impractical due to the high cost and human effort required for annotation. Consequently, real-world datasets frequently contain noisy labels, where the assigned labels are incorrect or misleading (Zhang et al., 2021a).

Noisy labels pose a significant challenge for DNNs because they tend to overfit the noise, leading to degraded generalization performance (Zhang et al., 2021a). The susceptibility of DNNs to memorize random labels (Zhang et al., 2021b) exacerbates this issue, making it crucial to develop methods that are robust to label noise. Existing approaches to address noisy labels can be broadly categorized into sample selection like MentorNet (Jiang et al., 2018) and Co-teaching (Han et al., 2020), which aim to identify clean samples from noisy datasets to train the model, reweighting techniques like the study by (Liu & Tao, 2015) which adjust the contribution of each sample during training, regularization methods like Nested Dropout (Chen et al., 2021) and Mixup (Zhang, 2017), and semi-supervised learning strategies like DivideMix (Li et al., 2020) which models loss distributions using Gaussian Mixture Models to distinguish between clean and noisy samples, treating the latter as unlabeled and applying SSL techniques.

Despite the progress, handling instance-dependent noise (IDN), where the probability of a label being noisy varies with the instance features, remains challenging (Yao et al., 2021). In real-world scenarios, noise is often instance-dependent, making it difficult to separate hard samples from mislabeled ones. Methods like CleanNet (Lee et al., 2018) learn to assess label correctness using a small clean validation set, while Pseudo-Label Correction (PLC) (Zhang et al., 2021c) refines labels by leveraging model predictions during training. LongReMix (Cordeiro et al., 2023) combines SSL with data augmentation to effectively handle IDN. However, most existing approaches primarily focus on

identifying or reweighting samples and ignore the potential of modeling the label corruption process itself to recover clean labels, which limits their capacity of disentangling complex noise patterns and reduces their effectiveness in practical applications.

In this paper, we propose Reciprocal Label Diffusion (RLD), a novel framework that leverages a mutual guidance mechanism between a label diffusion model and a prediction model to effectively learn from noisy labels. Diffusion models (Sohl-Dickstein et al., 2015; Ho et al., 2020) have emerged as powerful generative models capable of modeling complex data distributions. We harness this capability by modeling both the corruption and recovery processes of labels in the logit space. In RLD, the diffusion model is guided by the outputs of the prediction model to denoise corrupted labels through a forward and reverse diffusion process, thus modeling and correcting label noise with standard diffusion distributions while enforcing instance-dependency of the noise prediction. Subsequently, the prediction model is refined using the denoised labels produced by the diffusion model, enhancing its ability to learn accurate representations. This reciprocal interaction enables both models to iteratively enhance each other, effectively mitigating the adverse effects of noisy labels. Furthermore, we incorporate a contrastive denoising loss inspired by contrastive learning (Chen et al., 2020) to enforce consistency across different data augmentations. This loss enhances the model's robustness to noise and improves its discriminative capability. Our main contributions are summarized as follows:

- We introduce a novel framework, Reciprocal Label Diffusion (RLD), which leverages a mutual guidance mechanism between a label diffusion model and a prediction model to effectively learn from noisy labels.
- We propose a prediction guided label diffusion process in the logit space, where the diffusion model is guided by the output of the prediction model to denoise soft labels through their logits.
- We design a contrastive denoising loss to enforce consistency across different data augmentations, enhancing the diffusion model's robustness to label noise.
- We demonstrate through extensive experiments on benchmark datasets that our method outperforms existing state-of-the-art approaches, achieving substantial prediction improvements under various noise conditions.

## 2 RELATED WORK

### 2.1 LEARNING WITH NOISY LABELS

Deep neural networks (DNNs) are prone to overfitting noisy labels due to their capacity to memorize random data (Zhang et al., 2021a), leading to poor generalization on clean test sets. To tackle this issue, various methods have been proposed, which we categorize into general approaches for label noise and those specifically addressing instance-dependent noise (IDN).

**General Approaches for Learning with Label Noise**  One common strategy is sample selection, aiming to identify and utilize clean-label samples from noisy datasets. MentorNet (Jiang et al., 2018) trains a teacher network to provide curriculum guidance by selecting reliable samples based on loss values. Co-teaching (Han et al., 2020) involves two networks teaching each other by exchanging small-loss samples, reducing the impact of noisy labels. Decoupling (Malach & Shalev-Shwartz, 2017) updates model parameters only when two classifiers disagree on predictions, further mitigating noise effects. Reweighting methods constitute another category of techniques which adjust the influence of each sample during training. For example, (Liu & Tao, 2015) propose reweighting samples based on estimated noise rates, enhancing robustness. In addition, data augmentation techniques like Mixup (Zhang, 2017) blend input data and labels, regularizing models and diminishing sensitivity to label noise. Regularization approaches such as Nested Dropout (Chen et al., 2021) combine dropout with curriculum learning, gradually increasing sample difficulty to improve learning under noise. SELFIE (Song et al., 2019) iteratively identifies trustworthy samples and corrects mislabeled ones based on model confidence. Semi-supervised learning (SSL) has also been integrated to handle label noise effectively. In particular, DivideMix (Li et al., 2020) models loss distributions using Gaussian Mixture Models to distinguish between clean and noisy samples, treating the latter as unlabeled and applying SSL techniques.

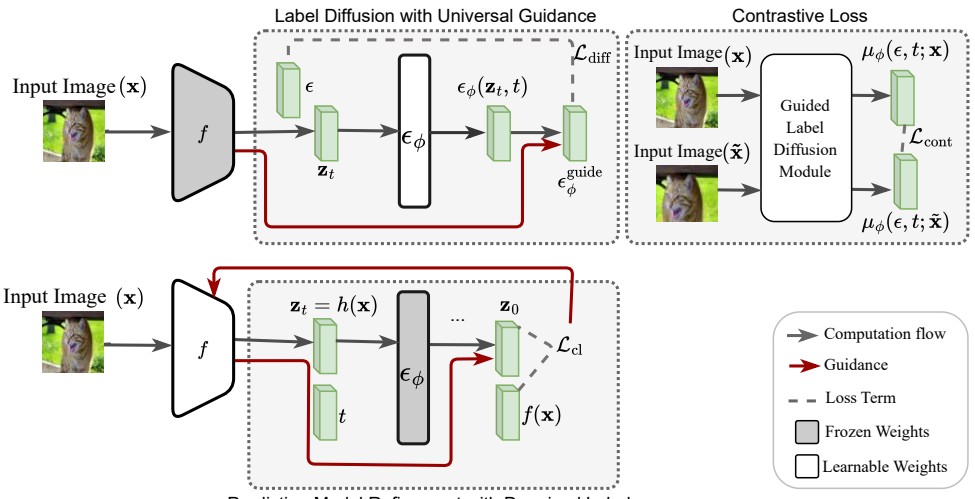

Figure 1: **Overview of the Reciprocal Label Diffusion (RLD) Framework.** The RLD framework consists of a universally guided label diffusion model $\epsilon_\phi$ and a label-denoising guided prediction model $f_\theta$, which iteratively refine each other through reciprocal learning. The diffusion model leverages universal guidance from the prediction model, aligning denoising with statistical accuracy using $\mathcal{L}_{\text{diff}}$, while the prediction model is updated with the denoised labels. The classification loss $\mathcal{L}_{\text{cl}}$ refines predictions based on denoised labels, establishing a synergistic feedback loop between models. A contrastive loss $\mathcal{L}_{\text{cont}}$ further strengthens robustness by enforcing consistency across augmentations.

**Methods for Instance-Dependent Label Noise**  In real-world scenarios, label noise is often instance-dependent, where the probability of a label being noisy varies with the instance features. This complexity makes it challenging to separate hard samples from mislabeled ones. (Yao et al., 2021) introduced a causal framework to model IDN but faced limitations due to reliance on co-teaching, which can accumulate errors when clean sample identification is not perfect. To overcome this, CleanNet (Lee et al., 2018) learns to assess label correctness using a small clean validation set, assisting in correcting noisy labels. Pseudo-Label Correction (PLC) (Zhang et al., 2021c) refines labels by leveraging model predictions during training, improving robustness to IDN. LongReMix (Cordeiro et al., 2023) extends SSL methods by combining consistency regularization with data augmentation to handle IDN effectively. Adaptive reweighting schemes like BARE (Patel & Sastry, 2023) adjust sample weights based on their likelihood of being clean, addressing cumulative errors in sample selection under IDN. In addition, SSR (Li et al., 2024) utilizes statistical estimations of noise rates to correct labels, enhancing performance.

## 2.2 Diffusion Models

**Diffusion Probabilistic Models**  Diffusion models have gained significant attention as a powerful class of generative models capable of producing high-quality data samples. Originally introduced by (Sohl-Dickstein et al., 2015), these models define a Markov chain that progressively adds Gaussian noise to data (forward process) and then learns to reverse this noising process to generate new data samples (reverse process). (Ho et al., 2020) revitalized diffusion models by proposing Denoising Diffusion Probabilistic Models (DDPM), which simplified the training objective and demonstrated state-of-the-art image generation results. (Nichol & Dhariwal, 2021) further improved upon DDPMs by optimizing the noise schedule and introducing variational bounds, enhancing sample quality and reducing training time. Score-based generative models, introduced by (Song & Ermon, 2019), unify diffusion models with score matching, enabling the use of stochastic differential equations to model the data generation process (Song et al., 2021). These advancements have established diffusion models as a leading approach in generative modeling, outperforming methods like Generative Adversarial Networks (GANs) (Dhariwal & Nichol, 2021).

**Guidance Techniques in Diffusion Models**  Guidance methods are crucial for controlling and improving the sample quality of diffusion models, especially for conditional generation tasks. Several

guidance techniques have been proposed to steer the generative process toward desired attributes. (Dhariwal & Nichol, 2021) proposed to integrate external classifiers as guidance. It uses the gradients from a pretrained classifier to adjust the denoising steps, effectively conditioning the generation on class labels. While this approach enhances sample fidelity and diversity, it requires training an additional classifier, increasing computational complexity. (Ho & Salimans, 2021) introduced classifier-free guidance. This technique trains the diffusion model jointly on conditional and unconditional objectives by randomly dropping conditioning information during training. At inference time, guidance is applied by extrapolating between the conditional and unconditional predictions, achieving conditional generation without a separate classifier and reducing computational overhead. Universal guidance (Bansal et al., 2024) leverages the denoised instance to guide the diffusion process, contrasting with classifier guidance that relies on the classification of the noisy instance. This approach potentially avoids suboptimal guidance, as classifiers pre-trained on clean instances may underperform when applied to noisy samples. Moreover, universal guidance extends the flexibility of guidance methods by allowing the integration of various guidance signals without retraining the diffusion model.

CARD treats labels as samples from a diffusion model, replacing the softmax head with a generative label denoiser (Han et al., 2022). We instead address noisy-label learning: a diffusion model denoises logits under universal guidance from a companion predictor, and the denoised outputs iteratively refine that predictor. This reciprocal loop (plus a contrastive denoising loss) corrects instance-dependent noise, unlike CARD's classifier replacement. LRA-Diffusion depends on CLIP-retrieved label prototypes, injecting strong external supervision (Chen et al., 2023). Our method uses no external knowledge, relying purely on reciprocal guidance between diffusion and prediction models, with a contrastive denoising loss for robustness.

## 3 METHOD

We consider the problem of learning prediction models from a classification dataset $\mathcal{D} = \{(\mathbf{x}_i, \widehat{\mathbf{y}}_i)\}_{i=1}^N$ with label noise, where $\mathbf{x}_i \in \mathcal{X}$ denotes an input sample and $\widehat{\mathbf{y}}_i \in \mathcal{Y}$ is the corresponding observed label vector, which may be a noisy version of the unknown true label vector $\mathbf{y}_i$. The goal is to learn a good prediction model $f_\theta : \mathcal{X} \to \mathcal{Y}$ parameterized by $\theta$ by simultaneously denoising the observed noisy labels.

In this section, we present our proposed approach, Reciprocal Label Diffusion (RLD), to address the challenge of learning from noisy labels through a novel diffusion-based framework, featuring reciprocal guidance between a diffusion model and a prediction model. The overall framework of RLD is illustrated in Figure 1. It consists of two main components: a universally guided label diffusion model and a label-denoising guided prediction model, which iteratively reinforce each other in a reciprocal learning process. Additionally, a novel contrastive diffusion loss is introduced to increase the diffusion model's robustness to noise. This loss encourages representations of different augmentations of the same input to align closely while distinguishing them from other inputs. The label diffusion model leverages outputs from the prediction model via a universal guidance mechanism, steering the denoising process toward statistically accurate labels based on the prediction model's current understanding of the data. In return, the prediction model is refined using the denoised labels generated by the diffusion model. The improved prediction model then provides enhanced guidance to the diffusion model in subsequent iterations. The RLD framework establishes a synergistic loop where the diffusion and prediction models guide each other, collaboratively denoising labels and refining predictions. Its details are elaborated in the following sections.

### 3.1 LABEL DIFFUSION WITH UNIVERSAL GUIDANCE

Diffusion models have been established as a leading approach for generating high quality data samples. A diffusion model comprises two primary processes: the forward process and the reverse process, each modeled as a Markov chain defined by Gaussian distributions. The forward process, i.e., the diffusion process, gradually adds Gaussian noise at each time step, starting from an initial sample that assumed to be clean, while the reverse diffusion process is an iterative denoising procedure that recovers clean samples.

In this approach, we propose to denoise the noisy labels and recover clean labels by learning a label diffusion model. In particular, we aim to denoise each observed label vector $\widehat{\mathbf{y}}_i$ to approximate the underlying true label vector $\mathbf{y}_i$. However, due to the probability distribution constraints in the label space such that each label vector should have nonnegative values that sum to 1, it is infeasible to directly deploy standard diffusion operations that rely on Gaussian distributions. To address this problem, our label diffusion model is designed to operate in the logit space of label vectors. To facilitate operations in this space, instead of directly using the observed label vectors $\{\widehat{\mathbf{y}}_i\}_{i=1}^N$ from the training data $\mathcal{D}$, we use the logits of the predicted soft label vectors from the prediction model $f_\theta$ initially trained on $\mathcal{D}$ as inputs for the label diffusion model. Specifically, the inputs in the logit space are produced by a logit function $h_\theta$, which is defined as the backbone part of the prediction model $f_\theta$ before the softmax activation:

$$\mathbf{z}_0 = h_\theta(\mathbf{x}) = \mathrm{logit}(f_\theta(\mathbf{x})) \tag{1}$$

where $\mathrm{logit}(\cdot)$ extracts the outputs of the linear prediction layer of $f_\theta$ before applying the softmax activation. In other word $f_\theta(\mathbf{x}) = \mathrm{softmax}(h_\theta(\mathbf{x}))$.

Modeling the diffusion process in the logit space maintains the standard diffusion operations, while preserving the relative relationships between classes, which is beneficial for classification tasks. This approach allows the model to learn how to recover clean labels from different noise levels, improving its robustness to label noise. To simulate varying levels of label noise, we define a forward diffusion process that progressively corrupts the input logits by adding Gaussian noise over $t$ time steps:

$$\mathbf{z}_t = \sqrt{\bar{\alpha}_t}\, h_\theta(\mathbf{x}) + \sqrt{1 - \bar{\alpha}_t}\, \boldsymbol{\epsilon}, \tag{2}$$

where $\alpha_t \in (0, 1)$ controls the variance of the noise at each step $t$ such that $\bar{\alpha}_t = \prod_{i=1}^t \alpha_i$ (Ho et al., 2020), and $\boldsymbol{\epsilon} \sim \mathcal{N}(\mathbf{0}, \mathbf{I})$ is Gaussian noise. This process progressively corrupts the logits, simulating different levels of label noise.

Following the standard diffusion models (Ho et al., 2020), a denoising network $\epsilon_\phi(\mathbf{z}_t, t)$ parameterized by $\phi$ can be learned to predict the noise added in $\mathbf{z}_t$ during the forward process, and deployed for reverse diffusion—i.e., label denoising. However, simply predicting the added noise is insufficient for the goal of supporting prediction model learning. To enforce the instance-dependency of the label diffusion process and incorporate the statistical prediction information, we propose to incorporate the outputs of the prediction model for label diffusion training as external guidance. Specifically, inspired by (Bansal et al., 2024), we propose to deploy a universal guidance that works on denoised label logits. The universal guidance function is defined as:

$$\ell\left(\widehat{\mathbf{y}}_0, f_\theta(\mathbf{x})\right) = \left\| \mathrm{softmax}(\widehat{\mathbf{z}}_0) - f_\theta(\mathbf{x}) \right\|^2, \tag{3}$$

where $\widehat{\mathbf{y}}_0 = \mathrm{softmax}(\widehat{\mathbf{z}}_0)$ denotes the estimated clean label vector computed from $\widehat{\mathbf{z}}_0$, while $\widehat{\mathbf{z}}_0$ is the estimated denoised logits at the time step 0 of the reverse diffusion process and is computed as:

$$\widehat{\mathbf{z}}_0 = \frac{1}{\sqrt{\bar{\alpha}_t}} \left( \mathbf{z}_t - \sqrt{1 - \bar{\alpha}_t}\, \epsilon_\phi(\mathbf{z}_t, t) \right). \tag{4}$$

Hence $\widehat{\mathbf{z}}_0$ as well as $\widehat{\mathbf{y}}_0$ can be treated as functions of $\mathbf{z}_t$.

In the reverse diffusion process of label diffusion, we modify the noise prediction function by incorporating the gradient of the universal guidance function with respect to the noisy logits $\mathbf{z}_t$ and obtain the following guided denoising function:

$$\epsilon_\phi^{\mathrm{guide}}(\boldsymbol{\epsilon}, t, \mathbf{x}) = \epsilon_\phi(\mathbf{z}_t, t) + s(t) \cdot \nabla_{\mathbf{z}_t} \ell\left(\widehat{\mathbf{y}}_0, f_\theta(\mathbf{x})\right), \tag{5}$$

where $s(t)$ is a time-dependent scaling factor controlling the strength of the guidance. By incorporating the gradient of the prediction based guidance, we steer the denoising process toward solutions that are not only plausible under the diffusion distributions but also better aligned with the current prediction model. This integration helps improve the correction of noisy labels by leveraging additional information from the input images $\{\mathbf{x}_i\}_{i=1}^N$.

To train the denoising network effectively, we deploy the following diffusion loss with the guided denoising predictions:

$$\mathcal{L}_{\mathrm{diff}}(\phi) = \mathbb{E}_{\mathbf{x} \in \mathcal{D}, t \sim [0:T], \boldsymbol{\epsilon} \sim \mathcal{N}(\mathbf{0}, \mathbf{I})} \left[ \left\| \epsilon_\phi^{\mathrm{guide}}(\boldsymbol{\epsilon}, t, \mathbf{x}) - \boldsymbol{\epsilon} \right\|^2 \right], \tag{6}$$

where $\boldsymbol{\epsilon}$ is the sampled target noise added during the forward diffusion process. This loss enforces the denoising network to be learned in an instance-dependent manner, encouraging the reverse label diffusion process to be statistically prediction consistent with the input image features.

## 3.2 ENHANCEMENT WITH CONTRASTIVE DIFFUSION LOSS

To further enhance the robustness and generalizability of the instance-dependent label diffusion model across various levels of noise and variations in the data, we devise a contrastive diffusion loss to enforce diffusion consistency across different variations of the same input instance $\mathbf{x}$.

In the reverse diffusion process, the conditional distribution of the logits at each time step $t$, $p_\phi(\mathbf{z}_{t-1}|\mathbf{z}_t)$, is modeled as a Gaussian distribution $\mathcal{N}(\mathbf{z}_{t-1}; \mu_\phi(\epsilon, t; \mathbf{x}), \sigma_t^2 \mathbf{I})$, where the mean vector $\mu_\phi(\epsilon, t; \mathbf{x})$ is computed through the guided denoising network $\phi$:

$$\mu_\phi(\epsilon, t; \mathbf{x}) = \frac{1}{\sqrt{\alpha_t}} \left( \mathbf{z}_t - \frac{1 - \alpha_t}{\sqrt{1 - \bar{\alpha}_t}} \epsilon_\phi^{\text{guide}}(\epsilon, t, \mathbf{x}) \right), \tag{7}$$

where, as previously introduced, $\bar{\alpha}_t = \prod_{i=1}^{t} \alpha_i$ and each $\alpha_i$ is a variance schedule hyperparameters at time step $t$ in standard diffusion models; $\mathbf{z}_t$ can be computed through Eq.(1). As the label denoising process in reverse diffusion is characterized by a sequence of denoising steps with $t \in \{T, \cdots 1\}$ defined by the Gaussian distributions above, it is critical to ensure the robustness of each denoising step.

To this end, we propose a contrastive diffusion loss to enhance the diffusion model by enforcing the mean vectors of the Gaussian distributions at each reverse diffusion time step $t$ to be relatively similar for variations of the same instance $\mathbf{x}$. Specifically, let $\tilde{\mathbf{x}}$ denote an augmented variation of $\mathbf{x}$. We define the contrastive diffusion loss as follows:

$$\mathcal{L}_{\text{cont}}(\phi) = - \mathbb{E}_{\mathbf{x}_i \in \mathcal{D}, t \sim [0:T], \epsilon \in \mathcal{N}(\mathbf{0}, \mathbf{I})} \left[ \log \left( \frac{\exp\left(\text{sim}_{\mu_\phi}(\epsilon, t, \mathbf{x}_i, \tilde{\mathbf{x}}_i))/\tau\right)}{\sum_{j=1}^{N} \mathbb{1}_{[i \neq j]} \exp\left(\text{sim}_{\mu_\phi}(\epsilon, t, \mathbf{x}_i, \mathbf{x}_j)/\tau\right)} \right) \right], \tag{8}$$

where $\tau$ is a temperature hyperparameter controlling the sharpness of the similarity distribution, and $\mathbb{1}_{[i \neq j]}$ is an indicator function equal to 1 when $i \neq j$ and 0 otherwise; $\text{sim}_{\mu_\phi}(\epsilon, t, \mathbf{x}, \tilde{\mathbf{x}})$ is a cosine similarity function defined as:

$$\text{sim}_{\mu_\phi}(\epsilon, t, \mathbf{x}, \tilde{\mathbf{x}}) = \text{cosine}\left(\mu_\phi(\epsilon, t; \mathbf{x}), \mu_\phi(\epsilon, t; \tilde{\mathbf{x}})\right). \tag{9}$$

By treating the denoised logits of labels as high-level representations of the corresponding data points, the intuition behind this contrastive loss is to encourage the representations for different views (or variations) of the same data point to be relatively similar, while distinguishing them from representations of other data points. This enables the model to better capture the intrinsic structure of the data in logit space, enhancing informative label denoising. From the label denoising perspective, this contrastive loss ensures that the universally guided reverse diffusion steps produce consistent outputs across different variations of the same input data points, while distinguishing them from those of other data points. This approach enforces label denoising to be both instance-dependency (i.e., instance-informative) and robust to random noise and variations.

## 3.3 PREDICTION MODEL REFINEMENT WITH DENOISED LABELS

Given a trained label diffusion model $\epsilon_\phi$, the noisy labels predicted from the current prediction model $f_\theta$ can be denoised through the reverse diffusion process. Specifically, we can start the reverse process at a random time step $t$ using the logits from the prediction model: $\mathbf{z}_t = h_\theta(\mathbf{x})$. Then from time step $t$ to time step 1, we iteratively denoise the logits using the guided denoising network $\epsilon_\phi^{\text{guide}}$, following the standard Markov decision process characterized by $p_\phi(\mathbf{z}_{t-1}|\mathbf{z}_t)$. Specifically, at each time-step $t$, we estimate the denoised logits for the next time step as the most likely sample—the mean of the Gaussian distribution of $p_\phi(\mathbf{z}_{t-1}|\mathbf{z}_t)$, such that:

$$\mathbf{z}_{t-1} = \frac{1}{\sqrt{\alpha_t}} \left( \mathbf{z}_t - \frac{1 - \alpha_t}{\sqrt{1 - \bar{\alpha}_t}} \epsilon_\phi^{\text{guide}}(\epsilon, t, \mathbf{x}) \right), \tag{10}$$

At the end of the reverse diffusion process, we compute the denoised label vector $\widehat{\mathbf{y}}_0$ from the denoised logits $\mathbf{z}_0$ at timestep zero by transforming the logits back to the probability simplex using the softmax function:

$$\widehat{\mathbf{y}}_0 = \text{softmax}\left(\mathbf{z}_0\right). \tag{11}$$

Table 1: Test accuracy and standard deviations (%) of different methods on CIFAR10-IDN and CIFAR100-IDN under various IDN noise rates. Columns indicate the label noise ratio.

| Method | IDN - CIFAR10 | | | | | IDN - CIFAR100 | | | | |
|---|---|---|---|---|---|---|---|---|---|---|
| | 0.20 | 0.30 | 0.40 | 0.45 | 0.50 | 0.20 | 0.30 | 0.40 | 0.45 | 0.50 |
| CE (Yao et al., 2021) | 75.8 | 69.2 | 62.5 | 51.7 | 39.4 | 30.4 | 24.2 | 21.5 | 15.2 | 14.4 |
| Mixup (Zhang, 2017) | 73.2 | 72.0 | 61.6 | 56.5 | 49.0 | 32.9 | 29.8 | 25.9 | 23.1 | 21.3 |
| Forward (Patrini et al., 2017) | 74.6 | 69.8 | 60.2 | 48.8 | 46.3 | 36.4 | 33.2 | 26.8 | 21.9 | 19.3 |
| Reweight (Liu & Tao, 2015) | 76.2 | 70.1 | 62.6 | 51.5 | 45.5 | 36.7 | 31.9 | 28.4 | 24.1 | 20.2 |
| Decoupling (Malach & Shalev-Shwartz, 2017) | 78.7 | 75.2 | 61.7 | 58.6 | 50.4 | 36.5 | 30.9 | 27.9 | 23.8 | 19.6 |
| Co-teaching (Han et al., 2020) | 81.0 | 78.6 | 73.4 | 71.6 | 45.9 | 38.0 | 33.4 | 28.0 | 25.6 | 24.0 |
| MentorNet (Jiang et al., 2018) | 81.0 | 77.2 | 71.8 | 66.2 | 47.9 | 38.9 | 34.2 | 31.9 | 27.5 | 24.2 |
| DivideMix (Li et al., 2020) | 94.8 | 94.6 | 94.5 | 94.1 | 93.0 | 77.1 | 76.3 | 70.8 | 57.8 | 58.6 |
| SSR (Feng et al., 2022) | 96.5 | 96.5 | 96.3 | 95.9 | 94.1 | 78.8 | 78.6 | 77.0 | 75.0 | 72.8 |
| kMEIDTM (Cheng et al., 2022) | 92.2 | 90.7 | 85.9 | - | 73.7 | 69.1 | 66.7 | 63.4 | - | 59.1 |
| InstanceGM Garg et al. (2023) | 96.6 | 96.5 | 96.3 | 96.1 | 95.9 | 79.6 | 79.2 | 78.4 | **77.4** | 77.1 |
| InstanceGM-E Garg et al. (2024) | - | - | - | - | - | 79.6 | 79.4 | 79.5 | - | **78.2** |
| HOC Zhu et al. (2021) | 90.0$_{(0.1)}$ | - | 85.4$_{(0.8)}$ | - | - | 67.4$_{(0.8)}$ | - | 61.2$_{(1.0)}$ | - | - |
| CC (Zhao et al., 2022) | 93.4$_{(0.1)}$ | - | 94.9$_{(0.0)}$ | - | - | 79.6$_{(0.1)}$ | - | 76.5$_{(0.2)}$ | - | - |
| RLD (Ours) | **97.9**$_{(0.1)}$ | **97.0**$_{(0.1)}$ | **96.8**$_{(0.1)}$ | **96.2**$_{(0.1)}$ | **96.0**$_{(0.2)}$ | **80.1**$_{(0.3)}$ | **80.0**$_{(0.3)}$ | **79.6**$_{(0.3)}$ | 77.0$_{(0.2)}$ | 75.3$_{(0.2)}$ |

For simplicity, we can encode this denoising process starting from a time step $t$ as a function $\widehat{\mathbf{y}}_0 = g_{\bar{\phi},\bar{\theta}}(\mathbf{x}, t)$, where the notation "$\bar{\cdot}$" denotes the stop-gradient operation, indicating that gradients are not back-propagated through the parameters $\phi$ and $\theta$ in this function.

By denoising the outputs of the current prediction model $f_\theta$ on the training data and using the denoised label vectors as the prediction targets, we can further refine the prediction model through the following classification loss:

$$\mathcal{L}_{\text{cl}}(\theta) = \mathbb{E}_{\mathbf{x}\in\mathcal{D}, t\sim[0:T]} \left[ \ell_{\text{CE}} \left( g_{\bar{\phi},\bar{\theta}}(\mathbf{x}, t), f_\theta(\mathbf{x}) \right) \right], \tag{12}$$

where $\ell_{\text{CE}}$ denotes the cross-entropy loss between the denoised label vector produced by $g_{\bar{\phi},\bar{\theta}}$ and the prediction model's current output $f_\theta(\mathbf{x})$. By further refining the prediction model with diffused prediction labels, we expect the prediction model can be improved to make more accurate predictions, which then provides better guidance to the diffusion model in subsequent iterations within the reciprocal learning framework. The use of stop-gradient ensures that during the optimization of $\theta$, gradients do not flow back through $\bar{\theta}$ and $\bar{\phi}$ in the diffusion process. This design isolates the update of $\theta$ based on the refined labels without affecting the parameters used in generating these labels, maintaining stability in the denoising and learning process.

### 3.4 RECIPROCAL LEARNING PROCESS

The overall learning problem of our RLD model is formulated as a minimization problem over the following training objective by incorporating the diffusion loss, the contrastive loss and the classification loss presented above:

$$\mathcal{L}_{\text{train}}(\phi, \theta) = \mathcal{L}_{\text{diff}}(\phi) + \lambda_{\text{cl}}\mathcal{L}_{\text{cl}}(\theta) + \lambda_{\text{cont}}\mathcal{L}_{\text{cont}}(\phi), \tag{13}$$

where $\lambda_{\text{cl}}$ and $\lambda_{\text{cont}}$ are hyperparameters controlling the trade-off between the losses.

To learn the RLD model, we first pretrain the probabilistic prediction model $f_\theta$ on the observed noisy data $\mathcal{D}$ by minimizing a standard cross-entropy loss, which provides the initial model parameters $\theta$ for the reciprocal learning process. Then we alternatively update the diffusion model $\phi$ and the prediction model $\theta$ by minimizing the objective function in Eq.(13) in a reciprocal manner. The reciprocal guidance mechanism allows the prediction model to be improved using the denoised labels from the diffusion model, while the diffusion model can also be further enhanced with the guidance provided by the refined prediction model.

## 4 EXPERIMENTS

### 4.1 EXPERIMENTAL SETUP

**Datasets** Our experiments utilize 5 distinct datasets. CIFAR10 and CIFAR100 (Krizhevsky et al., 2009) each comprise 50,000 training images and 10,000 test images. CIFAR10 is divided into 10 classes, while CIFAR100 also has 10 classes. Both datasets initially contain clean data. Following the methodology described by (Xia et al., 2020), we manually introduce instance-dependent label noise

Table 2: Test accuracy and standard deviations (%) of ANIMAL-10N. Bold values indicate the best performances.

| | CE+Dropout | SELFIE | PLC | Nested-CE | SSR+ | DISC | SURE | InstanceGM | RLD (Ours) |
|---|---|---|---|---|---|---|---|---|---|
| Accuracy | $81.3_{(0.3)}$ | $81.8_{(0.1)}$ | $83.4_{(0.4)}$ | $84.1_{(0.1)}$ | 88.5 | $87.1_{(0.1)}$ | 89.0 | 84.6 | $\mathbf{90.4}_{(0.1)}$ |

Table 3: Test accuracy and standard deviations (%) of Food-101N. Bold values indicate the best performances.

| | CleanNet | BARE | DeepSelf | PLC | LongReMix | DISC | SURE | RLD (Ours) |
|---|---|---|---|---|---|---|---|---|
| Accuracy | 83.9 | 84.1 | 85.1 | $85.2_{(0.0)}$ | 87.3 | 89.0 | 88.0 | $\mathbf{89.2}_{(0.2)}$ |

into the training sets. Animal-10N (Song et al., 2019) dataset is utilized as a benchmark containing ten classes of animals that are visually similar. It comprises a training set of 50,000 images and a test set of 5,000 images. The labels in the training set have an estimated noise ratio of 8%. Food-101N (Lee et al., 2018) dataset consists of 310,009 training images featuring various food recipes gathered from online sources, categorized into 101 classes. This dataset presents a label noise ratio of approximately 20%. Models trained on this dataset are evaluated using a clean-labeled test set from Food-101, which includes 25,250 images. Red Mini-ImageNet from CNWL (Jiang et al., 2020), comprises images and their corresponding labels sourced from the internet at various controllable label noise rates. This dataset is designed to examine real-world noise within a controlled environment. We select Red Mini-Imagenet for our analysis due to its representation of realistic label noise scenarios. The dataset includes 100 classes, each containing 600 images derived from the ImageNet dataset (Russakovsky et al., 2015). For consistency with prior studies (Xu et al., 2021), images are resized to $32 \times 32$ pixels, reduced from the original $84 \times 84$ pixels, and we explore noise rates of 20%, 40%, 60%, and 80%, aligning with the common configurations in the literature (Xu et al., 2021; Yao et al., 2021).

**Implementation Details** Following the previous studies (Cordeiro et al., 2023; Lv et al., 2022) we used a ResNet-34 for CIFAR10-IDN and a ResNet-50 for CIFAR100-IDN and Food-101N. For ANIMAL-10N, we use VGG-19 with batch normalization as in (Song et al., 2019) and PreAct ResNet-18 as backbone for Red Mini-Imagenet. In our study, the model was trained using stochastic gradient descent (SGD) with a momentum parameter set to 0.9, and a batch size of 128, complemented by L2 regularization at a coefficient of $5 \times 10^{-4}$. The model underwent training over 200 epochs for the CIFAR10, CIFAR100, Red Mini-Imagenet, and ANIMAL10N datasets. An initial learning rate of 0.01 was adjusted to 0.001 at the midpoint of the training epochs. The preliminary WarmUp phase varied across datasets, extending for 10 epochs in CIFAR10, and 30 epochs in CIFAR100, ANIMAL-10N, and Red Mini-Imagenet. In our implementation, we adopted the Latent Diffusion model (Rombach et al., 2022) maintaining all parameter settings consistent with those reported in the original study, and utilized the same architecture for the main classifier model as that used for the diffusion encoder. We use DDIM schedule with $T = 50$ denoising steps for all datasets. Specifically for for RLD we set $\lambda_{\text{cont}}, \lambda_{\text{cl}}, \tau$ to 0.5, 1, and 0.1 respectively.

## 4.2 COMPARISON RESULTS

We compare several methods, including CE (Yao et al., 2021), Mixup (Zhang, 2017), Reweight (Liu & Tao, 2015), Decoupling (Malach & Shalev-Shwartz, 2017), Co-teaching (Han et al., 2020), MentorNet, (Jiang et al., 2018), DivideMix (Li et al., 2020), SSR (Li et al., 2024), Nested-Dropout (Chen et al., 2021), CE+Dropout (Chen et al., 2021), SELFIE (Song et al., 2019), Nested-CE (Chen et al., 2021), CleanNet (Lee et al., 2018), BARE (Patel & Sastry, 2023), DeepSelf (Han et al., 2019), PLC (Zhang et al., 2021c), LongReMix (Cordeiro et al., 2023), DISK (Li et al., 2023), SURE (Li et al., 2024), HOC Zhu et al. (2021), InstanceGM Garg et al. (2023), InstanceGM-E Garg et al. (2024), using ResNet-18, ResNet-34, ResNet-50, VGG-9 as the backbone network.

Table 1 presents the comparative results on CIFAR10-IDN and CIFAR100-IDN datasets employing ResNet-34 and ResNet-50 as backbone networks, respectively. For CIFAR10-IDN, our methodology shows a significant 1.3% enhancement in performance over the second-best method, InstanceGM, at a noise level of 0.20. This improvement emphasizes the capability of our approach to sustain high accuracy despite increasing noise levels. For CIFAR100-IDN, RLD delivers superior or competitive performance across most noise levels, remaining on par with the strongest baselines where it does

Table 4: Test accuracy and standard deviations (%) of Red Mini-Imagenet. Columns indicate the label noise ratio. Bold values indicate the best performances.

| Method | 0.2 | 0.4 | 0.6 | 0.8 |
|---|---|---|---|---|
| CE (Yao et al., 2021) | 47.4 | 42.7 | 37.3 | 29.8 |
| Mixup (Zhang, 2017) | 49.1 | 46.4 | 40.6 | 33.6 |
| MentorMix (Jiang et al., 2020) | 51.0 | 47.1 | 43.8 | 33.5 |
| FaMUS (Xu et al., 2021) | 51.4 | 48.1 | 45.1 | 35.5 |
| DivideMix (Li et al., 2020) | 51.0 | 46.7 | 43.1 | 34.5 |
| SSR(Feng et al., 2022) | 52.2 | 49.0 | 42.4 | 33.2 |
| RLD (Ours) | $\mathbf{54.1}_{(0.3)}$ | $\mathbf{50.1}_{(0.4)}$ | $\mathbf{45.8}_{(0.6)}$ | $\mathbf{36.1}_{(0.9)}$ |

Table 5: Ablation study results in terms of test accuracy and standard deviations (%) results on IDN-CIFAR100. Columns indicate the label noise ratio. Bold values indicate the best performances.

| Method | 0.20 | 0.30 | 0.40 | 0.45 | 0.50 |
|---|---|---|---|---|---|
| RLD (Ours) | $\mathbf{80.1}_{(0.3)}$ | $\mathbf{80.0}_{(0.3)}$ | $\mathbf{79.6}_{(0.3)}$ | $\mathbf{77.0}_{(0.2)}$ | $\mathbf{75.3}_{(0.2)}$ |
| $-$w/o $\mathcal{L}_{\text{cont}}$ | $78.4_{(0.4)}$ | $77.9_{(0.1)}$ | $77.2_{(0.3)}$ | $76.3_{(0.2)}$ | $75.0_{(0.1)}$ |
| $-$w/o $\mathcal{L}_{\text{cl}}$ | $73.3_{(0.3)}$ | $75.5_{(0.2)}$ | $74.1_{(0.4)}$ | $75.1_{(0.3)}$ | $73.9_{(0.2)}$ |
| $-$ w/o Guidance | $77.9_{(0.2)}$ | $78.4_{(0.1)}$ | $77.0_{(0.2)}$ | $76.5_{(0.1)}$ | $74.1_{(0.3)}$ |

not strictly outperform them. This pattern shows that our approach scales well to more challenging, fine-grained settings while maintaining robustness under instance-dependent noise.

The results presented in Table 2 present the comparative results on the Animal-10N dataset employing VGG-9 as the backbone network. Our method attains a leading test accuracy of 90.4%, which represents a notable enhancement of 1.4% over the second best method, SSR. This illustrates the significant gains our approach offers over well-established methods in the field.

The results in Table 3 present the comparative results on the Food-10N dataset employing ResNet-50 as the backbone network. RLD achieves a test accuracy of 89.2%, outperforming all competing methods by a significant margin. Specifically, we observe an improvement of approximately 1.9% points over the previous best-performing method.

In Table 4, we report the comparative results on the Red Mini-Imagent dataset employing ResNet-18 as the backbone network. Our RLD proposed method RLD demonstrates superior performance over competing approaches on the Red Mini-Imagenet dataset across varying noise rates. Specifically, at a noise rate of 0.2, RLD improves test accuracy by approximately 1.9% compared to the second-best method, SSR. At a noise rate of 0.4, RLD surpasses SSR by about 1.2%, and at a noise rate of 0.6, it outperforms FaMUS by around 0.7%. These improvements highlight RLD's effectiveness in handling moderate to high levels of instance-dependent label noise through its reciprocal guidance mechanism between the diffusion and prediction models.

## 4.3 ABLATION STUDY

We conducted an ablation study to investigate the impact of different components in our proposed RLD framework on overall performance. The study focused on classification accuracy using the IDN-CIFAR100 dataset, with noise rates ranging from 0.20 to 0.50. As presented in Table 5, removing any key component from RLD leads to a noticeable decrease in performance across all noise levels. Specifically, we studied three variants: (1) '$-$w/o $\mathcal{L}_{\text{cont}}$" which removes the contrastive loss; (2) '$-$w/o $\mathcal{L}_{\text{cl}}$" drops fine-tuning of the prediction model with denoised labels; (3) $-$ w/o Guidance which removes the guidance mechanism in training diffusion model. When the contrastive loss is omitted, the accuracy drops noticeably compared to the full model, highlighting the importance of enforcing consistency across augmentations. Excluding the classification loss $\mathcal{L}_{\text{cl}}$ that fine-tunes the prediction model using denoised labels results in significant performance degradation, especially at higher noise rates, demonstrating the benefit of reciprocal guidance from the diffusion model. Lastly, removing the guidance mechanism also leads to lower accuracy, underscoring the critical role of mutual guidance These results confirm that each component contributes substantially to the robustness and effectiveness of RLD in handling noisy labels.

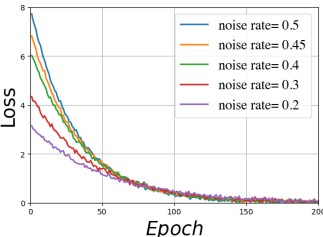

Figure 2: Training objective vs. epochs on CIFAR100-IDN at noise rates 0.2–0.5. RLD converges smoothly without oscillations.

### 4.4 CONVERGENCE ANALYSIS

As shown in Fig. 2, RLD exhibits smooth, non-oscillatory convergence on CIFAR100-IDN across all evaluated noise rates (0.2–0.5). The training objective decreases steadily throughout optimization, even at the highest noise level, without spikes or divergence, indicating robust and stable dynamics under substantial instance-dependent noise.

Importantly, the transition from the warm-up phase of the prediction model to the reciprocal learning phase does not introduce any visible instability: the curves remain smooth after the coupling between the diffusion model and the predictor is activated. We also observe that higher noise rates lead to slightly slower, but still monotonic, decrease of the objective, suggesting that RLD gracefully degrades in optimization speed without exhibiting collapse or mode switching. This behavior is consistent with our design choices, pretraining the predictor, using stop-gradient when forming denoised targets, and regularizing the diffusion process with contrastive consistency, which together prevent the two modules from amplifying each other's errors. Overall, the learning curves in Fig. 2 empirically support that the reciprocal training loop converges stably across a wide range of label-noise levels.

## 5 CONCLUSION

We introduced Reciprocal Label Diffusion (RLD), a novel framework that addresses learning from noisy labels through mutual guidance between a diffusion model and a prediction model. By modeling label corruption and recovery in the logit space, RLD effectively handles instance-dependent label noise. The diffusion model, guided by the prediction model's outputs, denoises corrupted labels, while the prediction model is refined using the denoised labels from the diffusion model. This reciprocal interaction enhances the robustness and accuracy of both models. Incorporating a contrastive denoising loss further improves model resilience by enforcing consistency across different data augmentations. Extensive experiments on benchmark datasets demonstrate that RLD outperforms state-of-the-art methods, achieving significant improvements in classification accuracy under various noise conditions.

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

---

**Algorithm 1** Reciprocal Label Diffusion (RLD)

---

**Input**: Noisy dataset $\mathcal{D} = \{(\mathbf{x}_i, \widehat{\mathbf{y}}_i)\}_{i=1}^N$, number of diffusion steps $T$, hyperparameters $\lambda_{\text{cl}}$, $\lambda_{\text{cont}}$, pre-trained prediction model $f_{\theta_0}$
**Output**: Trained $f_\theta$ and $\epsilon_\phi$
**for** iteration i = 1 **to** I **do**
    Compute initial logits using Eq. equation 1
    Sample $t \sim [0 : T], \epsilon \sim \mathcal{N}(0, \mathbf{I})$
    Compute corrupt logit $\mathbf{z}_t$ using Eq. equation 2
    Estimate denoised logits $\widehat{\mathbf{z}}_0$ using Eq. equation 4
    Compute guidance using Eq. equation 3
    Compute diffusion loss $\mathcal{L}_{\text{diff}}$ using Eq. equation 5
    Compute $\mu_\phi(\epsilon, t; \mathbf{x})$ for input samples augmentations using Eq. equation 7
    Compute contrastive loss $\mathcal{L}_{\text{cont}}$ using Eq. equation 8
    $\mathcal{L} = \mathcal{L}_{\text{diff}} + \lambda_{\text{cont}}\mathcal{L}_{\text{cont}}$
    Update diffusion model $\epsilon_\phi$ using gradient decent.
    Sample $t \sim [0 : T]$
    Compute noisy logits $\mathbf{z}_t = h_{\bar{\theta}}(\mathbf{x})$
    Compute $\widehat{\mathbf{y}}_0$ using Eq. equation 10 and Eq. equation 11.
    Compute classification loss $\mathcal{L}_{\text{cl}}$ using Eq. equation 12
    Update prediction model $f_\theta$ using gradient decent.
**end for**

---

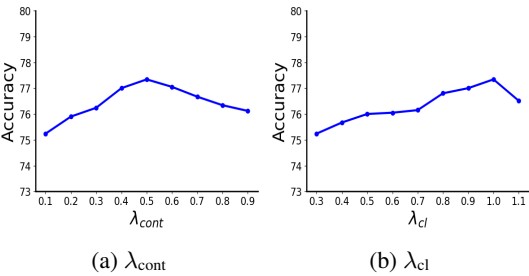

(a) $\lambda_{\text{cont}}$        (b) $\lambda_{\text{cl}}$

Figure 3: Sensitivity analysis for hyper-parameters $\lambda_{\text{cont}}$, $\lambda_{\text{cl}}$, on IDN-CIFAR100 with 0.50 noise rate.

## A   TRAINING ALGORITHM

The learning process of the proposed Reciprocal Label Diffusion (RLD) method is summarized in Algorithm 1.

## B   HYPER-PARAMETER SENSITIVITY ANALYSIS

To understand the impact of hyperparameters on the performance of our Reciprocal Label Diffusion (RLD) framework on the IDN-CIFAR100 dataset with a noise rate of 0.50. The two hyper-parameters investigated were: (1). $\lambda_{\text{cont}}$ the trade-off coefficient associated with the contrastive denoising loss $\mathcal{L}_{\text{cont}}$. (2).$\lambda_{\text{cl}}$ which control the magnitude of the classification loss. The experimental results for various hyper-parameter values are shown in Figure 3.

We observe a clear trend where the classification accuracy improves as $\lambda_{\text{cont}}$ increases from 0.1 to 0.5. This indicates that incorporating the contrastive denoising loss with a moderate weight enhances the model's robustness by effectively enforcing consistency across different data augmentations. However, when $\lambda_{\text{cont}}$ exceeds 0.5, a gradual decline in accuracy is observed. This suggests that overly emphasizing the contrastive loss may inadvertently overshadow the primary denoising and classification objectives, leading to suboptimal performance. Increasing $\lambda_{\text{cl}}$ from 0.3 to 1.0 leads to a consistent improvement in classification accuracy, reaching a peak at $\lambda_{\text{cl}} = 1.0$. This trend indicates that the classification loss $\mathcal{L}_{\text{cl}}$ contributes to refining the prediction. However, further increasing

$\lambda_{cl}$ beyond 1.0 results in a decline in performance. This suggests that excessively weighting the classification loss may cause the model to overfit the denoised labels.

