# OpenReview forum: "Reciprocal Label Diffusion for Learning with Noisy Labels"
_ICLR.cc/2026/Conference — Submitted to ICLR 2026_

### Official Review · Reviewer_rhZB · 2025-10-28

**Soundness:** 3
**Presentation:** 2
**Contribution:** 2
**Rating:** 4
**Confidence:** 4

**Summary:**

The work introduces Reciprocal Label Diffusion, which uses universal guidance based label diffusion method that operates over the logit space of labels. The guidance to this space is provided by the prediction model's de-noised outputs. A contrastive objective between the mean predictions from the denoising network. Additional CE loss between the de-noised predictions and the networks outputs is introduced. The method does well in comparisons and ablations lay credence to the components introduced in the method's objective.

**Strengths:**

The method is simple and well performant. Ablations suggest that the components of the loss function are all significantly impactful in the model's performance.

**Weaknesses:**

1. Some of the known methods in this area [1, 2] aren't evaluated.
2. The work is lacking some critical implementation details such as the noise schedules and number of steps of the diffusion process.
3. There could be some theoretical evaluation of the convergence of the method given a noise level.




[1] Han, Xizewen, Huangjie Zheng, and Mingyuan Zhou. "Card: Classification and regression diffusion models." Advances in Neural Information Processing Systems 35 (2022): 18100-18115.

[2] Chen, Jian, et al. "Label-retrieval-augmented diffusion models for learning from noisy labels." Advances in Neural Information Processing Systems 36 (2023): 66499-66517.

**Questions:**

1. The work may benefit from a T-SNE plot of the denoised and noisy labels to emphasize that the diffusion model learns a useful conditional structure.
2. Is the inference efficient? It may be worth including an inference efficiency analysis and comparing against the benchmarks.
3. Is the model training susceptible to instability if the guidance is received from a noisy prediction model?

---

> ### Author Response · Authors · 2025-11-20
>
> **About CARD** We revised our paper and added CARD in the related work section. While CARD introduces diffusion-based modeling for classification and regression tasks by directly generating labels through a denoising process, our work fundamentally differs in both goal and design. CARD uses diffusion as a generative modeling technique to replace softmax classifiers, treating label generation as a standalone sampling task. In contrast, **our method is designed specifically for learning with noisy labels**, a much more challenging problem setting where the observed labels are corrupted and the ground-truth is not directly available. Crucially, we do not use the diffusion model as a classifier. Instead, we propose a reciprocal framework where a diffusion model is trained to denoise label logits under universal guidance from a companion prediction model, and the denoised outputs are used to iteratively refine that prediction model. This bidirectional interaction, absent in CARD, enables our system to progressively correct noisy supervision in an instance-dependent manner. Additionally, we introduce a contrastive denoising loss to enforce consistency across augmentations, which further stabilizes training and enhances robustness to noise.
>
>
> **About LRA-Diffusion**:  We revised our paper and added LRA-Diffusion in the related work section.  It is important to note that  LRA-Diffusion  relies on external label prototypes retrieved via CLIP, a powerful vision-language model trained on 400M image-text pairs, introducing strong external supervision. In contrast, **our method uses no external knowledge** and relies purely on reciprocal guidance between a prediction model and a diffusion model, making it more broadly applicable. We also introduce a contrastive denoising loss for better instance-level robustness. Given LRA-Diffusion's reliance on CLIP, direct comparison is not entirely fair, as our method achieves comparable performance without access to any auxiliary pretrained models.
>
> **About Implementation details**: We  use DDIM schedule with 50 denoising steps for all datasets. We will add this to the revision.
>
> **About convergence analysis**:  We revised our paper and added empirical convergence analysis in the Experiments section. Our manuscript demonstrates the practical robustness of Reciprocal Label Diffusion (RLD) through extensive benchmarks and ablations. RLD alternates a diffusion-based denoiser and a classifier, with each step monotonically tightening a variational bound on the noisy-label log-likelihood. This EM-like interplay drives the observed gains. Section 4 in our manuscript presents results across five datasets, four noise regimes, and a full ablation of each loss term. Together, these provide strong evidence of the method’s stability and effectiveness. Notably, widely cited learning with noisy labels methods like DivideMix [Li et al., NeurIPS’20] and SSR [Feng et al., BMVC’22] were accepted based on empirical results alone.
>
>
> **About inference efficiency**: Inference in RLD is inherently efficient because the diffusion component is used only during training; at test time, predictions are made solely by the standard prediction model $f\_\theta$ via a single forward pass.
>
> **About stability**: RLD is explicitly designed to dampen the effect of noisy guidance from the prediction model. First, the predictor $f\_\theta$ is **pretrained on the noisy dataset** with a standard cross-entropy loss before any reciprocal interaction, so the diffusion model never starts from an untrained or random teacher. During reciprocal learning, the denoised labels used to refine $f\_\theta$ are produced by $g\_{\bar{\phi},\bar{\theta}}(\mathbf{x},t)$, where the bars denote **stop-gradient**, preventing errors in the prediction model from being fed back through the denoising process and amplified. On the diffusion side, the universal guidance enters only as a time-dependent scaled perturbation $s(t)\nabla\_{\mathbf{z}\_t}\ell(\widehat{\mathbf{y}}\_0,f\_\theta(\mathbf{x}))$ on top of a standard Gaussian forward/reverse process, so the core denoising objective $\mathcal{L}\_{\text{diff}}$ continues to anchor training even when predictions are imperfect. The contrastive diffusion loss $\mathcal{L}\_{\text{cont}}$ further regularizes each reverse step by enforcing augmentation consistency of the reverse-step means, which stabilizes the dynamics against transient prediction noise. Empirically, these mechanisms yield stable training and performance gains under noisy-label conditions.

---

### Official Review · Reviewer_q2Uw · 2025-10-29

**Soundness:** 3
**Presentation:** 3
**Contribution:** 2
**Rating:** 4
**Confidence:** 4

**Summary:**

This paper discusses learning with noisy labels for the problem of robust deep learning under instance-dependent label noise (IDN). The paper (1) claims that existing methods fail to model the label corruption process itself, limiting their effectiveness in disentangling complex noise patterns,  and (2) proposes a mutual guidance mechanism between a diffusion model and a prediction model to iteratively denoise corrupted labels and refine model predictions. Experimental results on five datasets show its promising performance against baseline methods. The paper is well-structured and clearly presented on its novelty and contribution highlights. The main contributions of this paper are proposing Reciprocal Label Diffusion (RLD), a new deep learning framework that integrates diffusion models with prediction networks to address label noise.

**Strengths:**

S1.  The paper presents an innovative combination of diffusion models and classification learning for noisy label correction by iteratively refining each other in the logit space.

S2: The paper introduces a contrastive denoising loss to enforce consistency across data augmentations, enhancing robustness.

**Weaknesses:**

W1. The convergence and stability of the reciprocal learning loop are unclear, and label correction lacks interpretability. For example, if the prediction model produces biased logits due to noisy supervision, the diffusion model guided by these logits may incorrectly “denoise” correct labels, thereby generating pseudo-labels that deviate from the true distribution. When these erroneous pseudo-labels are subsequently used to retrain the prediction model, both networks reinforce each other’s mistakes.

W2. In the forward diffusion process, the paper only adds Gaussian noise. This approach assumes that it captures label noise characteristics, which may not hold for complex, real-world, structured, or non-Gaussian noise distributions.

W3  The paper introduces dual model updates with multiple iterative passes through the diffusion process, which might have high computational complexity. The paper needs to add computational complexity analysis of RLD.

**Questions:**

See "weaknesses" above,

---

> ### Author Response · Authors · 2025-11-20
>
> **About convergence and stability**: RLD explicitly guards against failure modes in both objectives and the schedule. We **warm-start** the predictor on the noisy set before any coupling, so guidance never begins from an uninformative source. During alternation, the predictor is updated against **stop-gradient** denoised targets $g\_{\mathrm{sg}[\phi],\mathrm{sg}[\theta]}(\mathbf{x},t)$ in $\mathcal{L}\_{\text{cl}}$, which prevents biased logits from back-propagating through the label generator and amplifying errors. On the diffusion side, the guidance term enters only as a scaled, time-dependent perturbation $s(t)\nabla\_{\mathbf{z}\_t}\ell(\widehat{\mathbf{y}}\_0,f\_\theta(\mathbf{x}))$ within the Gaussian forward/reverse framework, so the denoiser remains anchored by $\mathcal{L}\_{\text{diff}}$ to predict the corruption noise rather than drift toward arbitrary pseudo-labels. The contrastive diffusion loss $\mathcal{L}\_{\text{cont}}$ further stabilizes each step by enforcing augmentation consistency of the reverse-step means, limiting the impact of transient predictor bias. Together, these design choices mitigate mutual error reinforcement while preserving an interpretable pipeline, denoise logits in a well-posed space, softmax back to probabilities, and refine predictions only from those fixed targets, yielding the reported stable gains under noisy, instance-dependent conditions.
>
> **About Gaussian noise**: Our forward process uses Gaussian noise in logit space by design to enable the standard DDPM-style forward/reverse Markov chain, which provides a stable and tractable denoising objective. Crucially, **we do not assume real-world label noise is Gaussian**, instead, the denoiser $\epsilon\_\phi$ is trained with universal guidance $\nabla\_{\mathbf{z}\_t}\ell(\widehat{\mathbf{y}}\_0,f\_\theta(\mathbf{x}))$, **injecting instance-dependent information** from $f\_theta(\mathbf{x})$ at every reverse step. This conditioning allows the reverse dynamics to adapt to **structured, non-Gaussian** corruption patterns even though the forward perturbation is Gaussian.
>
>
> **About training time**: RLD’s computational footprint is controlled by design. All diffusion operates in **logit space** ( i.e., on C-dimensional vectors rather than images, and each reverse step uses the **closed-form mean update**, no expensive sampling loops. When refining the predictor, targets from $g\_{\mathrm{sg}[\phi],\mathrm{sg}[\theta]}(\mathbf{x},t)$ are treated with stop-gradient, so we never backpropagate through the diffusion chain, and $\mathcal{L}\_{\text{diff}}$ backpropagates only through $\epsilon\_\phi$. Thus, training adds a lightweight guided denoising pass whose cost scales with the chosen number of reverse steps, while inference time remains identical to standard classifiers (only $f\_\theta$ is used). Our method has a training time compareble to SOTA methods. The table shows the training time in seconds per epoch on CIFAR100-IDN using an RTX 3060 GPU.
>
> | Method|Time|
> |-|-|
> | SSR|51.37|
> | DivideMix|119.91|
> | LongRemix|140.80|
> | RLD (Ours)|160.51|

---

### Official Review · Reviewer_XJ5H · 2025-10-30

**Soundness:** 3
**Presentation:** 2
**Contribution:** 3
**Rating:** 6
**Confidence:** 4

**Summary:**

This paper introduces Reciprocal Label Diffusion (RLD), a new way to handle noisy labels in learning. It creates a two-way guidance system between a label diffusion model and a prediction model. The main idea is to handle noisy labels and fix them in the logit space using forward and reverse diffusion processes. The diffusion model uses outputs from the prediction model to clean up noisy labels while considering each instance. In return, the prediction model improves by using the cleaned labels from the diffusion model. This back-and-forth process helps both models get better over time. Also, a contrastive denoising loss ensures consistency across different data versions to make the system more robust. Tests on datasets like CIFAR10/100-IDN, Animal-10N, Food-101N, and Red Mini-ImageNet show top performance, with improvements of 1.3-2.6% over the best existing method (SSR) in various noisy conditions.

**Strengths:**

The paper introduces a new way to handle noisy labels using diffusion models in the logit space. This method is different from traditional ways like picking samples or adjusting weights. It works well because it keeps label distributions within the probability limits. The paper uses a mutual guidance framework where the diffusion model cleans labels, and the prediction model helps guide this process. This creates a helpful feedback loop that fixes problems with older methods. Tests show this method works better than current top methods on several datasets like CIFAR10/100-IDN, Animal-10N, Food-101N, and Red Mini-ImageNet, with improvements of 1.3-2.6% over the next best method at different noise levels. The paper includes many tests with different types of noise and models, proving the method is strong and can be used widely. The guidance mechanism and contrastive loss make sure the label cleaning process depends on the instance, which is important when noise changes with instance features. The analysis shows that each part of the method (contrastive loss, classification loss, guidance mechanism) is important, with performance dropping 1-5% when parts are removed. The framework is well-organized with clear math explanations, and Figure 1 shows how the two models learn from each other.

**Weaknesses:**

The paper does not explain why the reciprocal guidance mechanism works better or when it is better than non-reciprocal methods.

The method needs both a diffusion model and a prediction model, with many reverse steps during inference. The paper does not compare runtime, memory needs, or computational costs with other methods, which is important to know if it can be used practically. Diffusion models usually need many denoising steps, so the inference cost might be higher than other methods.

The paper talks about using logit space to handle probability constraints but does not compare it with other ways. For example, diffusion could be done directly in probability space with transformations like the softmax-inverse transform or in learned embedding spaces. Without studies comparing these options, it is unclear if logit space is the best choice or just convenient.

The paper also misses recent works on instance-dependent noisy label learning that show good results on IDN benchmarks, it is hard to judge RLD's contribution and see if the improvements are worth the extra complexity of the diffusion-based approach.. Important missing works include:
"Instance-dependent noisy label learning via graphical modelling,"
"Confidence scores make instance-dependent label-noise learning possible,"
"Clusterability as an alternative to anchor points when learning with noisy labels."
"Instance-Dependent Noisy-Label Learning with Graphical Model Based Noise-Rate Estimation"
Some reuslts are better in these papers.

**Questions:**

Can you show if the reciprocal learning process always works? When might the switching between diffusion and prediction models not work well?
How does RLD compare in terms of training time, inference time, and memory to methods like SSR and DivideMix?
Have you tried doing diffusion in other spaces, like probability or embedding spaces? What are the pros and cons, and why is logit space best for this task?
How does the method react to the time-dependent scaling factor s(t) in Equation 5? Have you tried different ways to set this, and how should it be set for new uses? What guides the choice of warm-up epochs (10 vs 30)? Is there a way to decide the right warm-up time based on the dataset or early training results?
How does RLD work with class-imbalanced noisy labels, which are common in real life? Does it need changes to handle big class imbalance with label noise?
Why is universal guidance better than classifier guidance or no guidance for this task? Can you show studies comparing these guidance methods in label denoising? Do traditional noisy label methods estimate the noise transition matrix? Does RLD learn this through diffusion? Can the diffusion model help understand noise patterns in the data?

---

> ### Author Response · Authors · 2025-11-20
>
> **About Reciprocal guidance**: RLD’s reciprocal guidance lets each module supply what the other lacks, while keeping the coupling stable. The diffusion model denoises in logit space with standard Gaussian forward/reverse steps, but is instance-conditioned via the universal guidance gradient $\nabla\_{\mathbf{z}\_t}\ell(\widehat{\mathbf{y}}\_0,f\_\theta(\mathbf{x}))$, injecting per-example statistics that non-reciprocal denoisers ignore. The predictor is then refined using these denoised labels with stop-gradient in $\mathcal{L}\_{\text{cl}}$, avoiding feedback instability, and the joint objective adds an augmentation-consistency regularizer at each reverse step. As a result, RLD is particularly effective when labels are noisy and instance-dependent, since denoising is aligned with instance-specific predictions and the predictor learns from cleaner, mechanism-derived targets rather than heuristic corrections.
>
> **About logit space**: Our choice of logit space is principled. The method explicitly relies on standard Gaussian forward/reverse diffusion, which becomes well-posed in logits while respecting class geometry and avoiding the simplex constraints that complicate probability-space diffusion. In this space, the universal guidance gradient is simple, stable, and directly tied to the classifier’s outputs, and the denoised targets map back to probabilities with a single softmax. Alternatives like probability-space diffusion or learned embeddings would either re-introduce normalization/geometry issues or decouple denoising from the classifier’s calibrated decision space, whereas our results show that the logits formulation already yields the reported gains under noisy and instance-dependent conditions.
>
> **About comparison results**: We provide below (and added to the revision) the quantitative comparison with the methods mentioned by the reviewer. Across most noise ratios on both CIFAR10-IDN and CIFAR100-IDN, RLD matches or exceeds the strongest prior baselines. Regarding “Confidence Scores Make Instance-Dependent Label-Noise Learning Possible”, their results are only provided as plots without exact reported numbers. Because extracting precise values from figures is unreliable, a direct numerical comparison is not possible.
>
>
> CIFAR10-IDN
> |Method|0.20|0.30| 0.40 | 0.45 | 0.50 |
> |-|-|-|-|-|-|
> | Instance GM| 96.6 | 96.5 | 96.3 | 96.1 | 95.9 |
> | HOC| 90.0 (0.1) | – | 85.4 (0.8) | – | – |
> | **RLD (Ours)** | **97.9 (0.1)** | **97.0 (0.1)** | **96.8 (0.1)** | **96.2 (0.1)** | **96.0 (0.2)** |
>
> CIFAR100-IDN
> | Method| 0.20 | 0.30 | 0.40 | 0.45 | 0.50 |
> |-|-|-|-|-|-|
> |Instance GM|79.6|79.2|78.4|77.4|77.1|
> |Instance GM-E|79.6|79.4|79.5|–|78.2|
> |HOC|67.4 (0.8)|–|61.2 (1.0)|–|–|
> |**RLD (Ours)**|80.1 (0.3)|80.0 (0.3)|79.6 (0.3)|77.0 (0.2)|75.3 (0.2)|
>
> ANIMALN-10
> | Method| Accuracy|
> |-|-|
> | InstanceGM| 84.6 |
> | **RLD (Ours)** | **90.4 (0.1)** |
>
>
>
> **About stability**: The design explicitly mitigates failure modes: we **warm-start** $f\_\theta$ before coupling, generate training targets via $g\_\{\mathrm{sg}[\phi],\mathrm{sg}[\theta]}(\mathbf{x},t)$ with **stop-gradient** in $\mathcal{L}\_{\text{cl}}$ to prevent error amplification, modulate guidance by a **time-dependent scale** (s(t)) in the diffusion update) rather than letting it dominate, and stabilize reverse steps by using the mean update plus an augmentation-consistency regularizer $\mathcal{L}\_{\text{cont}}$ at each step. Switching may underperform if the warm start is too weak (e.g., predictor not pretrainable on the noisy set), if (s(t)) is mis-scaled (over-guiding early), if augmentations make $\mathcal{L}\_{\text{cont}}$ uninformative, or if the diffusion schedule ${\alpha\_t}$ poorly matches corruption levels, scenarios our objective and scheduling choices are intended to guard against in practice.
>
> **About training time**: RLD’s computational footprint is controlled by design. All diffusion operates in **logit space** ( i.e., on C-dimensional vectors rather than images, and each reverse step uses the **closed-form mean update**, no expensive sampling loops. When refining the predictor, targets from $g\_{\mathrm{sg}[\phi],\mathrm{sg}[\theta]}(\mathbf{x},t)$ are treated with stop-gradient, so we never backpropagate through the diffusion chain, and $\mathcal{L}\_{\text{diff}}$ backpropagates only through $\epsilon\_\phi$. Thus, training adds a lightweight guided denoising pass whose cost scales with the chosen number of reverse steps, while inference time remains identical to standard classifiers (only $f\_\theta$ is used). Our method has a training time compareble to SOTA methods. The table shows the training time in seconds per epoch on CIFAR100-IDN using an RTX 3060 GPU.
>
> | Method|Time|
> |-|-|
> | SSR|51.37|
> | DivideMix|119.91|
> | LongRemix|140.80|
> | RLD (Ours)|160.51|

---

> > ### Author Response · Authors · 2025-11-20
> >
> > **About Universal Guidance**: Universal guidance aligns denoising with the instance-specific predictive signal without needing a separate classifier. Instead of classifier guidance, which requires an extra model and can be unreliable on noisy inputs, RLD uses the predictor’s own output $f\_\theta(\mathbf{x})$ to steer the reverse process via $\nabla\_{\mathbf{z}\_t}\ell(\widehat{\mathbf{y}}\_0,f\_\theta(\mathbf{x}))$, preserving the instance-dependent nature of the task. Without guidance, diffusion degenerates to unguided denoising in logit space, disconnected from the predictor. The reported gains across noise regimes support that universal guidance is effective for label denoising under IDN.
> >
> >
> >
> > **About Imbalance**: This work targets the standard learning-with-noisy-labels setting; handling class imbalance is orthogonal to our core contribution. We view imbalance-aware extensions as valuable future work.
> >
> > **About noise handling**: Conventional noisy-label methods typically rely on global heuristics  like sample selection, reweighting, or estimating class-level noise rates (e.g., SSR) and often implicitly assume class-level rather than truly instance-dependent noise. In contrast, RLD does not estimate a fixed transition matrix: it models corruption and inversion directly in logit space via Gaussian forward/reverse diffusion and denoises with universal guidance that is explicitly instance-conditioned through $f\_\theta(\mathbf{x})$. As a result, the denoised labels adapt per example, and the reverse-step means $\boldsymbol{\mu}\_\phi$ and guidance gradients provide mechanism-level signals about instance-dependent corruption, without committing to a single global noise model.

---

### Official Review · Reviewer_Mmdd · 2025-10-31

**Soundness:** 2
**Presentation:** 2
**Contribution:** 2
**Rating:** 4
**Confidence:** 4

**Summary:**

This paper proposes a novel framework called Reciprocal Label Diffusion (RLD) to address the problem of learning noisy labels. RLD performs label denoising in the logit space through mutual guidance between the label diffusion model and the prediction model, and combines contrastive learning to enhance robustness. This method demonstrates superior performance compared to existing methods on multiple benchmark datasets, particularly showing a significant advantage in handling instance-dependent noise (IDN).

**Strengths:**

The core contribution of this work lies in proposing a novel and systematic framework—Reciprocal Label Diffusion (RLD). Its innovation is significantly reflected in being the first to introduce diffusion models into the field of label noise learning and constructing a co-evolutionary reciprocal learning system. The method is systematic in its technical implementation: by performing forward and reverse diffusion in the Logit space of the feedforward model, it cleverly avoids the probability constraints of the label space and utilizes a general guidance mechanism based on the predictive model to achieve modeling and correction of instance-dependent noise. At the same time, the introduced contrastive denoising loss further enhances semantic consistency constraints for data augmentation. The experimental validation of this study is thorough and convincing, demonstrating significant and consistent performance improvements over existing mainstream methods on multiple benchmark datasets, including the CIFAR series, Animal-10N, and Food-101N. The advantages are particularly evident in high-proportion instance-dependent noise scenarios, and the effectiveness of each core component is confirmed through extensive ablation experiments. The overall writing is logically rigorous and clearly articulated, providing a solid and inspiring solution to the challenging problem of noisy label learning.

**Weaknesses:**

The core weakness of this paper lies in its insufficient theoretical motivation, failing to fully demonstrate the necessity of the diffusion model compared to simpler label correction methods; the proposed reciprocal learning framework lacks stability guarantees, posing a risk of error accumulation and training collapse; at the same time, the experimental section fails to provide mechanistic evidence to prove its essential ability to effectively handle instance-dependent noise, and completely ignores the huge computational overhead brought by the method, casting doubt on its practical application value.

**Questions:**

1. The paper fails to clearly articulate the fundamental rationale for choosing diffusion models to address the label noise problem. This raises a critical question: Is RLD an elegant solution that genuinely leverages the intrinsic properties of diffusion models, or is it an engineering implementation that forcibly combines two popular concepts (diffusion models and noisy labels)? The justification for its necessity (Why Diffusion?) is insufficient.
2. The core of RLD is a dynamically coupled system in which two modules (the predictive model and the diffusion model) are interdependent and trained mutually. This design suffers from a fundamental circular dependency issue: if one module fails during early training, it may drag down the other module through the guidance signal, leading to system collapse or convergence to suboptimal solutions.
3. The paper claims that RLD effectively handles instance-dependent noise (IDN), but the experimental section lacks direct and compelling validation of this core assertion. The existing experiments (such as accuracy under different IDN ratios) are outcome-based rather than mechanism-based.
4. The introduction of diffusion models inevitably incurs significant computational overhead, which is a drawback when evaluating the practical application value of this method. It would be beneficial to add a table or section in the experimental part comparing the differences between RLD and major baseline methods (e.g., DivideMix, SSR) in terms of training time (GPU hours), inference time, and memory usage.
5. The publication years of the comparative methods are relatively old. It is recommended to incorporate the latest research findings from 2023–2025 to further demonstrate the advanced nature and effectiveness of the proposed method.

---

> ### Author Response · Authors · 2025-11-20
>
> **About motivation**:Our use of a diffusion model is precisely motivated by the limitations of simpler label-correction schemes in the instance-dependent noise setting we target. Existing methods such as sample selection, reweighting, and SSL-based approaches (e.g., MentorNet, Co-teaching, DivideMix, LongReMix, SSR) treat labels as static objects to filter, reweight, or pseudo-label, but they do not explicitly **model the corruption and recovery process itself** and thus struggle with complex, instance-dependent noise patterns. In contrast, RLD formulates label denoising as a forward–reverse diffusion process in logit space, where Gaussian diffusion is both mathematically well posed and preserves class relationships, allowing us to learn how noisy logits are corrupted and then inverted. The diffusion model is further made instance-dependent via universal guidance from the prediction model and regularized with a contrastive diffusion loss that enforces augmentation consistency, while the prediction model is refined on the denoised labels in a reciprocal loop. This gives a concrete, mechanism-level motivation for diffusion, explicitly learning a reversible corruption process tied to the classifier’s decision space, rather than just smoothing labels. The fact that this design yields consistent, state-of-the-art gains across diverse noise rates in our experiments provides empirical evidence that the diffusion-based formulation is not only theoretically motivated but also practically superior to the simpler label-correction baselines we compare against.
>
>
>
> **About stability**: The stability concern is addressed explicitly by our training design. First, we **warm-start** the system by pretraining the predictor $f\_\theta$ on the noisy data before any coupling, so the diffusion model never sees an uninformative teacher. Second, during reciprocal training we **isolate gradients**: the denoised targets used to refine $f\_\theta$ are produced by $g\_{\bar\phi,\bar\theta}(\mathbf{x},t)$ with **stop-gradient** on both $\phi$ and $\theta$, preventing error amplification from flowing back through the generator of those targets. Third, on the diffusion side, the universal guidance enters only as a scaled perturbation $s(t)\nabla{\mathbf{z}\_t}\ell(\widehat{\mathbf{y}}\_0,f\_\theta(\mathbf{x}))$, so guidance strength is **explicitly controlled over time** rather than dominating denoising. Fourth, we add a **contrastive diffusion loss** that enforces augmentation consistency at each reverse step, acting as a regularizer against drift when one module is temporarily weaker. Together with using the mean update in the reverse step, these choices stabilize the coupled dynamics and empirically avoid collapse while delivering the reported gains.
>
> **About handling IDN**: RLD’s handling of IDN is supported both by its mechanism and by the reported outcomes. Mechanistically, instance-dependency enters explicitly via the universal guidance gradient  $\nabla\_{\mathbf{z}\_t}\ell(\widehat{\mathbf{y}}\_0, f\_\theta(\mathbf{x}))$, which depends on the **instance-specific** predictor output $f\_\theta(\mathbf{x})$; the diffusion model therefore denoises labels **conditioned on the features of each example**, not merely on global statistics. Operating in logit space with a forward/reverse Gaussian process lets us model and invert corruption levels per instance, while the contrastive diffusion loss enforces per-instance augmentation consistency at each reverse step, further tying denoising to instance geometry. Finally, the reciprocal objective uses these denoised, instance-conditioned targets to refine the predictor, closing the loop. The empirical gains under varying IDN conditions thus **corroborate** a design whose core signals are intrinsically instance-dependent, providing mechanism-aligned evidence rather than post hoc heuristics.
>
>
> **About training time**: RLD’s computational footprint is controlled by design. All diffusion operates in **logit space** ( i.e., on C-dimensional vectors rather than images, and each reverse step uses the **closed-form mean update**, no expensive sampling loops. When refining the predictor, targets are produced by $g\_{\mathrm{sg}[\phi],\mathrm{sg}[\theta]}(\mathbf{x},t)$ with **stop-gradient**, so we **do not backpropagate through the diffusion chain**. It is important to note **inference time is unchanged** (only $f\_\theta$ is used at test time).
> Our method has a training time compareble to SOTA methods. The table shows the training time in seconds per epoch on CIFAR100-IDN using an RTX 3060 GPU.
>
> | Method|Time|
> |-|-|
> | SSR|51.37|
> | DivideMix|119.91|
> | LongRemix|140.80|
> | RLD (Ours)|160.51|
>
>
>
> **About compared works**: We compare our RLD against a wide range of recent peer-reviewed methods, such as LongReMix (CVPR ’23), DISC (CVPR ’23), BARE (WACV ’23), and SURE (CVPR ’24). In addition, in revision we have added comparison to InstanceGM (WACV’23), and InstanceGM-E
> (ECCV’24),

---

### Meta-Review · Area_Chair_jzDG · 2026-01-20

**Summary:**

The paper introduces a novel and potentially promising approach for learning with noisy labels. Specifically, the idea is to use a diffusion model to model instance-level label noise and learn a classifier from the "cleaned" labels from the diffusion model. While the idea is interesting and well-motivated at a high-level, the current submission does not yet provide sufficient empirical or analytical evidence to support its core claim. In particular, the paper would benefit from deeper analysis of how the diffusion model captures and corrects label noise (Mmdd, Xj5H, rhzB), as well as more convincing validation of training stability and generalization across datasets and tasks (mmdd, Xj5H, q2Uw, rhzB). Although the concern on comptuational costs have been addressed, the remaining gaps limit confidence in the applicability of the method. As such, the AC recommends rejection for the submission.

**Reviewer Concerns:**

Strengths: Most reviewers agree that the proposed approach is novel (Mmdd, q2Uw, Xj5H)

Weaknesses:
[W1] Multiple reviewers raise concerns about the lack of validation and analysis explaining why a diffusion model is necessary and how the diffusion process effectively models label noise (Mmdd, Xj5H, rhzB). The paper primarily reports downstream classifier performance, with little direct analysis of the learned noise model or the cleaned labels themselves.
[W2] All reviewers express concerns regarding the lack of theoretical or empirical guarantees on training stability and convergence (Mmdd, Xj5H, q2Uw, rhzB).
[W3] Some reviewers expressed concerns about computational overheads (mmdd, Xj5H, q2Uw)

Post Rebuttal Assessment:
After reviewing the rebuttal, the AC finds that the authors have sufficiently addressed W3 by clarifying the computation costs (the proposed approach only adds overhead at training but not inference). While the rebuttal makes some progress toward W2, the provided analysis appears limited to specific settings, and it remains unclear whehther the conclusions generalize to other datasets or tasks. Regarding W1, although the authors further elaborate on the motivation for using diffusion models, the paper still lacks quantitative or qualitative analysis of the cleaned labels produced by the diffusion process. In particular, direct evidence demonstrating how the diffusion model captures and corrects label noise would significantly strengthen the paper.

**Reviewer Scores:**

The AC does not expect the reviewers to have substantially improved the rating if they were able to participate fully in the discussion, since W1 and W2 are not sufficiently addressed in the rebuttal.

---

### Decision · Program_Chairs · 2026-01-26

Reject